# Improvement of Lithium Storage Performance of Silica Anode by Using Ketjen Black as Functional Conductive Agent

**DOI:** 10.3390/nano12040692

**Published:** 2022-02-19

**Authors:** Guobin Hu, Xiaohui Sun, Huigen Liu, Yaya Xu, Lei Liao, Donglei Guo, Xianming Liu, Aimiao Qin

**Affiliations:** 1Key Lab New Processing Technology for Nonferrous Metal and Materials Ministry of Education, College of Material Science and Engineering, Guilin University of Technology, Guilin 541004, China; kokpinhu@hotmail.com (G.H.); 15564222794@139.com (X.S.); huigenliugz@163.com (H.L.); xuyayachonga@163.com (Y.X.); 2Guangxi Key Laboratory of Environment Pollution Control Theory and Technology, College of Environmental Science and Engineering, Guilin University of Technology, Guilin 541004, China; gdl0594@163.com; 3Key Laboratory of Function-Oriented Porous Materials, College of Chemistry and Chemical Engineering, Luoyang Normal University, Luoyang 471934, China; myclxm@163.com

**Keywords:** silica-based anode, Ketjen Black, electrochemical properties, lithium-ion battery

## Abstract

In this paper, SiO_2_ aerogels were prepared by a sol–gel method. Using Ketjen Black (KB), Super P (SP) and Acetylene Black (AB) as a conductive agent, respectively, the effects of the structure and morphology of the three conductive agents on the electrochemical performance of SiO_2_ gel anode were systematically investigated and compared. The results show that KB provides far better cycling and rate performance than SP and AB for SiO_2_ anode electrodes, with a reversible specific capacity of 351.4 mA h g^−1^ at 0.2 A g^−1^ after 200 cycles and a stable 311.7 mA h g^−1^ at 1.0 A g^−1^ after 500 cycles. The enhanced mechanism of the lithium storage performance of SiO_2_-KB anode was also proposed.

## 1. Introduction

Lithium-ion batteries (LIBs) have attracted much attention due to their high energy density and long cycle life. To meet the demand for scaled-up LIBs, the development of electrode materials with high performance is necessary. Graphite is widely used as anode material for LIBs [1], however, its theoretical lithium storage capacity is relatively low, only 372 mA h g^−1^. Therefore, silicon-based anode materials with higher theoretical specific capacity (4200 mA h g^−1^) are considered to be anode materials for next-generation LIBs [2,3,4]. However, the severe volume expansion (>300%) associated with the various phase transitions during the intercalation and escape of lithium in/out Si particles have been a major disadvantage, as this led to rapid capacity fading and significantly limited commercial application [5]. Although novel silicon anodes with nanosphere [6,7], nanotube [8,9], core-shell structure [10,11] and other new structures could improve the cycling performance [12,13], the complicated process and expensive preparation technology are prohibitive. In addition, the low initial coulombic efficiency and the poor conductivity also limited its application [14,15,16]. Compared to elemental Si anode, silicon oxides show a smaller volume change during cycling. Furthermore, when using silicon oxides as anodes, the in situ generated Li_2_O and lithium silicates during the first lithiation may buffer the large volume change and lead to the improvement of cycling stability. In cutting-edge researches, silica [17] with hollow [18], porous [19,20], and other special structures were composited with carbon [19,21,22], graphite [23], metal [24], and metal oxides [25,26,27,28] to improve its conductivity and lithium storage performance. These methods could effectively improve the electrochemical performance of silica anodes.

Carbon conductive agents, which were added during the electrode manufacturing process, played an important role in the impedance and electrode density. However, their functional mechanism still needs further investigation [29,30]. The literature has reported that conductive agents could be used as the mediator [31] to form a conductive network in electrodes, reducing the contact resistance of the electrode and improving the electron transport rate. Commercial carbon black, such as acetylene black (AB) and Super-P (SP), have been used as conductive agents in LIBs [32,33]. Compared with AB and SP, Ketjen Black (KB) has the advantages of large specific surface area, excellent electrical conductivity, and relatively narrow pore size distribution, when used as the conductive agent [34]. However, the systematic study of the effect of KB on silica anodes is sparsely reported.

Herein, a network nanostructure of silica (SiO_2_) anode material using KB as a conductive agent with high electrochemical performance was prepared. The effects of KB on the electrochemical performance of silica anode materials were systematically studied. Furthermore, the enhanced storage mechanism of the SiO_2_-KB anode materials was proposed. This work revealed that the type of conductive agent played a key role on the electrochemical performance of anode materials.

## 2. Materials and Methods

### 2.1. Synthesis

Briefly, SiO_2_ aerogels were prepared by the sol–gel method [35]. It was obtained by taking 8 mL of anhydrous ethanol in a beaker, adding ammonia to adjust the pH to 9–10, then slowly adding 0.5 mL of TEOS, and left for 4 h at room temperature before adding 1 mL of deionized water to prepare the gel, and freeze-drying to obtain SiO_2_ aerogels.

SiO_2_ nanospheres were obtained by first taking 3 mL of NH_3_·H_2_O and 60 mL of alcohol to be mixed and stirred thoroughly, then 1.5 mL of TEOS was added into the above solution and continued stirring for 10 h at room temperature to obtain a white emulsion, finally the solid product was collected by centrifugation, washed several times with distilled water and alcohol, and dried at 70 °C for 12 h in a vacuum.

### 2.2. Materials Characterization

Morphological and compositional analyses for the as-prepared sample were performed with Transmission Electron Microscope (TEM, JEM-2100F, JEOL Inc., Tokyo, Japan) and field emission scanning electron microscopy (SEM, S-4800, HITACHI Inc., Tokyo, Japan), respectively, the crystallographic structure of the obtained SiO_2_ were characterized by X-ray diffraction (XRD, X’ Pert PRO, PANalytical Inc., Almelo, The Netherlands), the chemical component of the SiO_2_ anode was investigated using an X-ray photoelectron spectroscope (XPS, ESCALAB 250Xi, Thermo Fisher Scientific Inc., Waltham, MA, USA) using Al kα radiation, the electrical resistance and electrical resistivity were tested by four-point probe meter (FPM, RTS-2A, 4 PROBES TECH Inc., China).

### 2.3. Electrochemical Measurements

The working electrodes were fabricated by compressing a mixture of the active materials (SiO_2_ nanospheres), a conductive material of KB, AB, or SP, and a binder of polyvinylidene fluoride at the mass ratio of 50:30:20 onto Cu foil current collector (10 μm in thickness), then dried at 110 °C for 12 h. 0.1 mL of 1 M LiPF_6_ in EC/DMC/DEC (Ethylene carbonate/Dimethyl carbonate/Diethyl carbonate) with volume ratio of 1:1:1 was used as the electrolyte, electrochemical experiments of half cells were carried out in CR2025 coin-type cells, 0.6 mm thick lithium discs are used as counter electrodes, and Polypropylene diaphragm type Celgard 2500 as battery separator. The cells were assembled in an argon-filled glove box (MIKROUNA, LAB2000, Shanghai, China). The specific capacity was measured by a galvanostatic discharge–charge method in the voltage range between 3.0 V and 0.01 V at a current density of 100 mA g^−1^ with SiO_2_ as the active material mass on a battery test system (Neware, BTS 5 V 10 mA, Shenzhen, China). Cyclic voltammetry was performed using an electrochemical workstation (CV, CHI 690D, CH Instruments Ins, Wuhan, China) between 3 V and 0.01 V (vs. Li/Li^+^) at a scan rate of 0.5 mV s^−1^.

## 3. Results and Discussion

Figure 1a shows the SEM image of the SiO_2_ aerogel, and it can be seen that the prepared SiO_2_ aerogel particles are uniform in size with sphere in shape. Figure 1b shows that the average size of SiO_2_ aerogel particle is about 100 nm. Figure 1c is the XRD pattern of the SiO_2_ aerogel, the crystal structure of the prepared silica aerogel only shows a broad diffraction peak at around 23°, indicating an amorphous structure.

Figure 2 shows the TEM and elemental mapping images of SiO_2_ aerogel mixed with different conductive agents, respectively. It can be seen that different conductive agents form different structures when combined with silica aerogel. Figure 2a shows that the aerogel mixture with KB (SiO_2_-KB) has an internet structure with uniformly dispersed nanoparticles, which indicates that a conductive network can be formed to provide a large number of effective conductive pathways and contacts for Li-ions. Furthermore, the elemental distribution shows a cobweb-like carbon chain pathway. For comparison, we also investigated the differences in the composition of the conductive pathways of SiO_2_-SP and SiO_2_-AB, respectively, under identical conditions. Figure 2b shows that the SiO_2_-SP has a branched structure with larger SiO_2_ particles and more agglomerates than that of SiO_2_-KB. Figure 2c shows that the SiO_2_-AB stacks together and has more agglomerate structure than that of SiO_2_-SP and SiO_2_-KB. Therefore, it is clear that the SiO_2_-KB has the best dispersion, indicating it has excellent conductive network channels.

To further confirm the effects of the three conductive agents on the electrochemical performance of SiO_2_ anode, a four-probe electrical resistance test was carried out and the result is shown in Table 1, the type of conductive agents plays an important role on the electrical resistance of SiO_2_ anode. KB provides much lower electrical resistance and electrical resistivity than SP and AB for SiO_2_ anode, which is helpful to improve the rate performance of electrode.

The cycling performance and coulombic efficiency of the SiO_2_ anode with different conductive agents are shown in Figure 3a. The first discharge capacity of SiO_2_-KB reaches 378.2 mA h g^−1^, the capacity has a slight decrease in the several consequent cycles, and maintains 351.4 mA h g^−1^ after 200th cycles at 0.2 A g^−1^. In comparison to SiO_2_-KB, electrodes of SiO_2_-SP and SiO_2_-AB exhibit a lower reversible capacity of 139.4 mA h g^−1^ and 118.7 mA h g^−1^ at the first cycle and after 100th cycles display the capacity of 163.9 mA h g^−1^ and 137.7 mA h g^−1^ at 0.2 A g^−1^, respectively, which indicates that KB is more beneficial in facilitating the silica electrochemical reaction. The specific capacity of SiO_2_-KB decreased before the first 40 cycles, and then gradually increased, even after more than 100 cycles; the former is mainly due to the gradual lithiation of SiO_2_ and the generated irreversible products such as lithium silicate and Li_2_O, and the latter is due to the generated elemental silicon, which can provide the reversible specific capacity by the Si-Li alloy reaction.

The rate capability of the electrodes is shown in Figure 3b, with different current densities of 0.2, 0.4, 0.6, 0.8, 1.0, 2.0, and 4.0 A g^−1^. The SiO_2_-KB electrode attains an average discharge capacity of 318.2 mA h g^−1^, 263.6 mA h g^−1^, 231.9 mA h g^−1^, 213.4 mA h g^−1^, 207.7 mA h g^−1^, 176.7 mA h g^−1^, and 151.4 mA h g^−1^, respectively, at the above current densities. The stable high reversible capacity of 331.1 mA h g^−1^ recovered when the current density turned back to 0.2 A g^−1^. Compared to SiO_2_-SP and SiO_2_-AB, SiO_2_-KB shows more excellent rate properties, indicating that the SiO_2_-KB has excellent structural stability.

To determine the origin of the electrochemical behavior of SiO_2_-KB, electrochemical impedance spectra (EIS) test was performed at their open-circuit potential. The equivalent circuits inserted in Figure 3c were employed to analyze the Nyquist plot of the desired anode material. The total impedance could be regarded as the electrolyte resistance R_e_ and the charge transfer resistance R_ct_, and C_dl_ is the double-layer capacitance. Z_w_ is the Warburg impedance that reflects the diffusion of lithium-ion in the solid. C_L_ means the simplified intercalation capacitance. A semicircle was an indication for the charge transfer at high frequency range, while the straight line for the low frequency lithium-ion diffusion in the electrode material [36]. Obviously, the resistance of SiO_2_-KB (R_ct_ = 130 Ω) is much lower than that of SiO_2_-SP (R_ct_ = 180 Ω) and SiO_2_-AB (R_ct_ = 240 Ω), suggesting that KB could remarkably enhance the silica electrical conductivity. Furthermore, SiO_2_-KB presents an exciting long-term cycling performance and delivers a specific capacity of 311.7 mA h g^−1^ at a current density of 1.0 A g^−1^ after 500 cycles. In comparison to SiO_2_-KB, SiO_2_-SP and SiO_2_-AB exhibit a much lower reversible capacity of 66.4 mA h g^−1^ and 75.9 mA h g^−1^, respectively, at the first cycle, and after 500th cycles, the capacity only remains 115.5 mA h g^−1^ and 123.9 mA h g^−1^, respectively, which is shown in Figure 3d.

To confirm the structural integrity of the electrodes after cycle tests, SEM images of the different electrodes with the three conductive agents after 200 cycles were obtained and illustrated in Figure 4. For the electrode of SiO_2_-KB, the shape of SiO_2_ remained constant after 200 cycles (Figure 4a,b). For the electrode of SiO_2_-SP, the silica undergoes a slight agglomeration phenomenon after 200 cycles (Figure 4c,d), while for the electrode of SiO_2_-AB, the SiO_2_ particles stick together and form large particles after 200 cycles (Figure 4e,f), which reduces the contact area between the silica, and leads to the reduction in capacity.

Figure 5 shows the CV curves of the SiO_2_ aerogel electrodes at a scan rate of 0.5 mV s^−1^ with three different conductive agents, respectively. As can be seen in Figure 5, the reduction characteristic peak potential for the reaction of silica to produce lithium silicate and lithium oxide is 0.42 V, 0.65 V, and 0.8 V when using KB, SP, and AB as the conductive agent, respectively. The peak potential is significantly shifted to a smaller voltage for the KB (Figure 5a) compared to the SP (Figure 5b) and AB (Figure 5c). The oxidation characteristic peak potential of 0.25 V for the silicon-lithium alloy when using KB as the conductive agent also shows a significant shift to a smaller voltage compared to the SP (0.28 V) and AB (0.35 V), which is mainly due to the fact that KB has higher conductivity than SP and AB, where KB acts as a microcurrent collector between SiO_2_ and the current collector to accelerate the speed of electron movement and also effectively increase the migration rate of Li^+^ in the electrode material. Subsequently, the polarization of the silica anode is reduced, which will facilitate the occurrence of electrochemical reactions [37].

The chemical states of Si in SiO_2_ anode with KB (Figure 6a), SP (Figure 6b) and AB (Figure 6c) during discharge/charge were identified by XPS, where energy correction for surface contamination was performed using C1s (284.6 eV) as a standard. The Si 2p_3/2_ peak shifts from 104.0 to 103.0 eV when discharged to 0.01 V, suggesting the reduction in Si to Li_x_Si. When charged to 3 V, the peak shifts back to the original position of the SiO_2_ electrode before discharge/charge (blue curve). It is clear that the curve in SiO_2_-KB fluctuates more strongly than the smooth curves in SiO_2_-SP and SiO_2_-AB, indicating that the electrochemical reaction promoted by SiO_2_ using KB as a conductive agent produces a higher amount of products containing elemental silicon, which is conducive to the improvement of electrochemical performance.

Based on previous researches, the possible electrochemical reaction mechanisms of SiO_2_ can be summarized into the following reactions [38,39]:(R1)$${\mathrm{SiO}}_{2}{+4\mathrm{Li}}^{+}{+4e}^{-}\to {2\mathrm{Li}}_{2}O+\mathrm{Si}$$
(R2)$${2\mathrm{SiO}}_{2}{+4\mathrm{Li}}^{+}{+4e}^{-}\to {\mathrm{Li}}_{4}{\mathrm{SiO}}_{4}+\mathrm{Si}$$
(R3)$${\mathrm{Si}+\mathrm{xLi}}^{+}{+\mathrm{xe}}^{-}↔{\mathrm{Li}}_{x}\mathrm{Si}$$

Reactions of R1, and R2 are irreversible (potential of KB-0.25 V, SP-0.28 V, AB-0.35 V) and occur simultaneously although they compete with each other, and the obtained Si is electrochemically active, while Li_2_O, and Li_4_SiO_4_ are electrochemically inactive [22]. The reaction R1 can produce more Si and result in a higher capacity than the reaction R2 [13].

To better understand the storage mechanism and excellent high-rate performance of the SiO_2_ aerogel electrode with KB, the CV curves of the SiO_2_ aerogel electrode under the three carbon conductive agents at different scan rates from 0.5 to 5 mV s^−1^ were collected and shown in Figure 7a,e,i. Generally, the current obeys a function relationship with the voltage during the sweep [40,41,42,43]:(1)$$i=av^{b}$$
where a and b are the parameters. The capacity contributions from the diffusion-controlled intercalation process and the surface-induced capacitive process can be qualitatively analyzed by the b-value. For a diffusion-controlled process, the b-value is 0.5, while the b-value near 1 means a totally capacitive-controlled process. According to the fitted line log (v)-log (i) curve depicted in Figure 7b,f,j, the b-value is 0.84, 0.86, 0.66, respectively. Furthermore, in view of the capacity contribution, the current (i) under a certain potential (V) can be divided into two parts [42,43,44]:(2)i(V)=k1v+k2v12
where $k_{1}v$ and k2v12 present charge stemmed from the surface capacitive charge and diffusion-controlled charge, respectively. The area share of the pseudocapacitive behavior of the silica aerogel electrodes with the three different conductive agents is shown in the paler parts of Figure 7c,g,k. The pseudocapacitance contributions are shown in Figure 7d,h,l. The pseudocapacitance contribution of the SiO_2_ aerogel electrode with KB at scan rates of 1.0 to 5.0 mV s^−1^ is 41%, 49%, 54%, 58% and 61%, respectively; 51%, 58%, 63%, 67% and 70%, respectively, for SP and 65%, 73%, 77%, 79% and 80%, respectively, for AB. It can be seen that the use of different forms of conductive agents has a greater effect on the contribution of the pseudocapacitance in the SiO_2_ aerogel electrode. The storage mechanism of the SiO_2_-KB electrode is dominated by diffusion-controlled intercalation behavior, which is due to the excellent conductive network structure of KB and leads to an accelerated redox reaction. The contribution of surface-driven pseudocapacitance behavior for the SiO_2_-KB electrode gradually increases as the scan rate increases, but is still less than that SiO_2_-SP and SiO_2_-AB. The contribution of the pseudocapacitance behavior of the SiO_2_-KB electrode increases with increasing scan rate, however remains smaller than that of the SiO_2_-SP and SiO_2_-AB electrodes.

Li^+^ diffusion coefficients during electrochemical charge/discharge for silica aerogels with different conductive agents calculated from the GITT method [45,46,47] are presented in Figure 8. The voltage change curve of the first charge/discharge under pulse current when using three different carbon conductive agents is shown in Figure 8a, and the Li-ion diffusion coefficient calculated from the pulse charge/discharge curve is shown in Figure 8b. The longer charging and discharging duration of Li^+^ in the diffusivity test curve of SiO_2_-KB is due to the special network structure of KB, which enables the nano-SiO_2_ particles to perform electrochemical reactions without agglomeration and less hindering to the transport of Li^+^. This indicates that the KB conductive agent accelerates the diffusion of Li^+^ and enables the sufficient electrochemical reaction of SiO_2_ [16].

In order to confirm the effect of the KB, the electrochemical properties of SiO_2_ nanosphere anode was compared with three different conductive agents, respectively. The morphology of the SiO_2_ nanospheres is shown in Figure 9a, which has a nice monodispersity and smooth surface with an average particle size of 100 nm. The inset in Figure 9a shows the XRD pattern of the SiO_2_ nanospheres; a broad peak at 23° suggesting that the SiO_2_ nanospheres are amorphous in structure similar to the SiO_2_ aerogels. Figure 9b–d show the CV curves of the SiO_2_ nanosphere anode at a scan rate of 0.5 mV s^−1^ when using the three different conductive agents, respectively. The reduction characteristic peak potential for the reaction of SiO_2_ to produce Li_2_O and Li_4_SiO_4_ locates at 0.5 V for the SiO_2_ nanospheres-KB anode, which has a significant shift to the left compared to the anodes of SiO_2_ nanospheres-SP (0.85 V) and SiO_2_ nanospheres-AB (0.9 V), indicating that it also has the similar effect of reducing polarization of SiO_2_ anodes prepared by different methods when using KB as the conducting agent, and strongly promotes the electrochemical reaction.

The function of KB can be described as a schematic diagram conductive network, as shown in Figure 10, the special structure of KB connects the SiO_2_ aerogel particles together and forms an excellent conductive network of dispersed SiO_2_-KB, which provides rich electron transfer channels and improves the electrochemical performance of silica anode.

## 4. Conclusions

In summary, the effects of the types and the structures of conductive agents on the electrochemical performance of SiO_2_ aerogel electrode were investigated, and the results show that the KB as a conductive agent not only can uniformly disperse and wrap the SiO_2_ nanoparticles, but also can build a good conductive network to enhance the transport rate of lithium-ions and effectively increase their electrochemical activity. This work proves and verifies that SiO_2_ aerogel can be used as a recommended electrode material for high-rate LIBs through choosing appropriate conductive agent.

## Figures and Tables

**Figure 1 nanomaterials-12-00692-f001:**
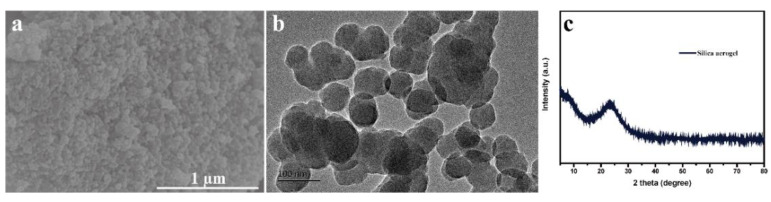
SEM (**a**), TEM (**b**) image, and XRD (**c**) pattern of silica aerogel.

**Figure 2 nanomaterials-12-00692-f002:**
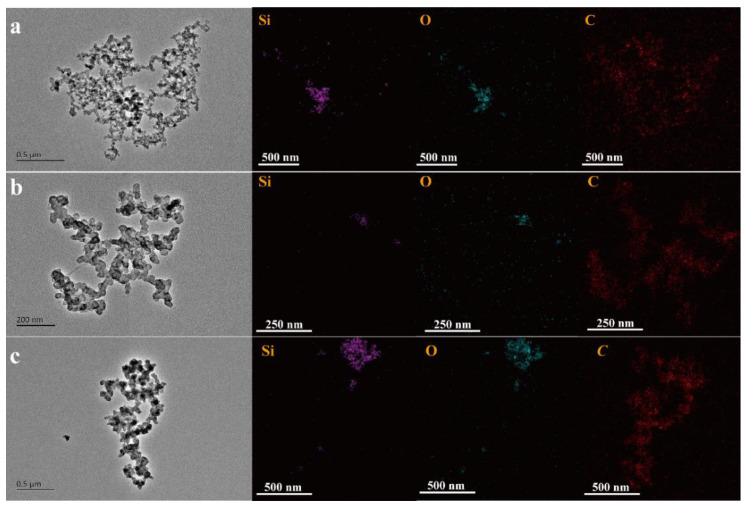
TEM and elemental mapping images of silica aerogels mixed with KB (**a**), SP (**b**), and AB (**c**).

**Figure 3 nanomaterials-12-00692-f003:**
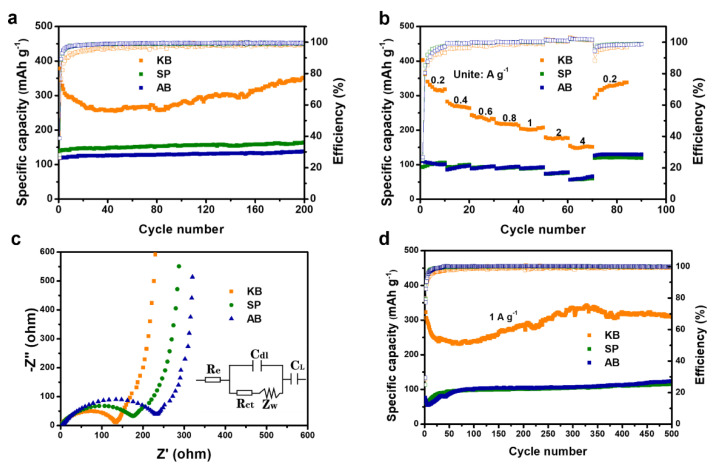
Cycle performance with coulombic efficiency at a current density of 200 mA g^−1^ (**a**), Rate capability at various current densities (**b**), Nyquist plot (**c**), and long-term cycling performance at a current density of 1.0 A g^−1^ (**d**) of silica aerogel anode with three carbon conductive agent.

**Figure 4 nanomaterials-12-00692-f004:**
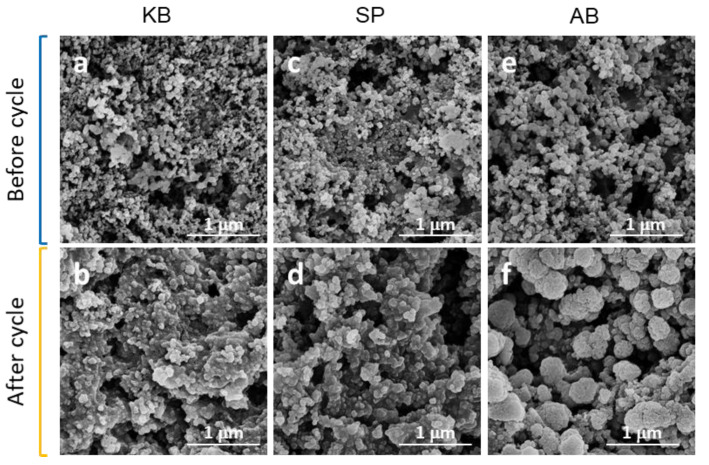
SEM images of SiO_2_ electrode with KB (**a**,**b**), SP (**c**,**d**), and AB (**e**,**f**) before and after 200 cycles.

**Figure 5 nanomaterials-12-00692-f005:**
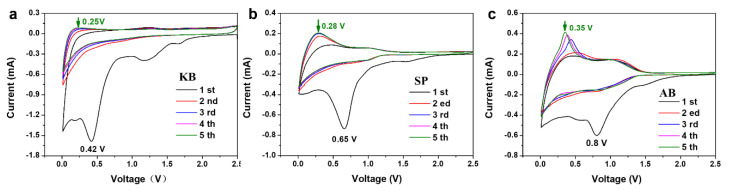
CV curve for the first five cycles of the silica aerogel when KB (**a**), SP (**b**), and AB (**c**) as conductive agents.

**Figure 6 nanomaterials-12-00692-f006:**
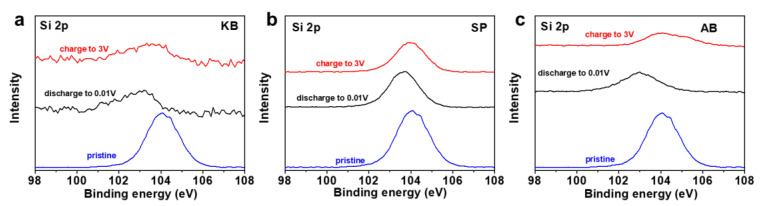
Si 2p XPS spectra of SiO_2_ anode with KB (**a**), SP (**b**) and AB (**c**) at different states.

**Figure 7 nanomaterials-12-00692-f007:**
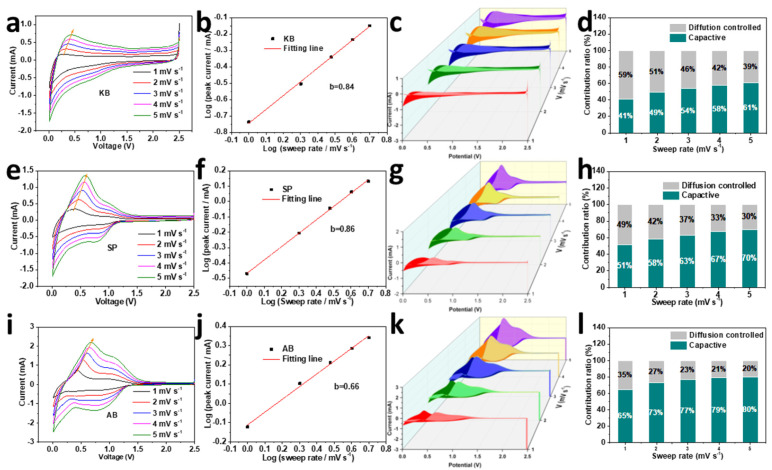
Behaviors of pseudocapacitance in silica aerogels using KB (**a**–**d**), SP (**e**–**h**), and AB (**i**–**l**) as conductive agents.

**Figure 8 nanomaterials-12-00692-f008:**
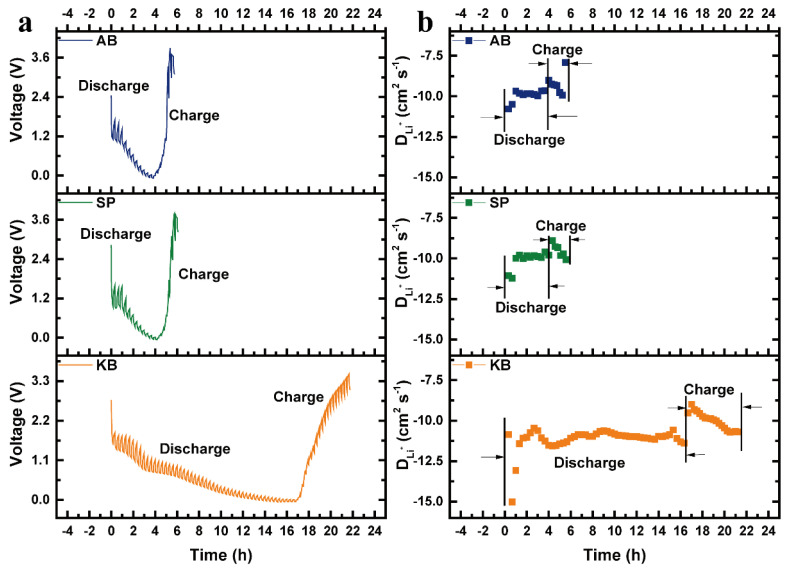
The voltage change curve of the first charge/discharge under pulse current (**a**) and Li-ion diffusion coefficient calculated from the pulse charge/discharge curve (**b**) with different carbon conductive agents.

**Figure 9 nanomaterials-12-00692-f009:**
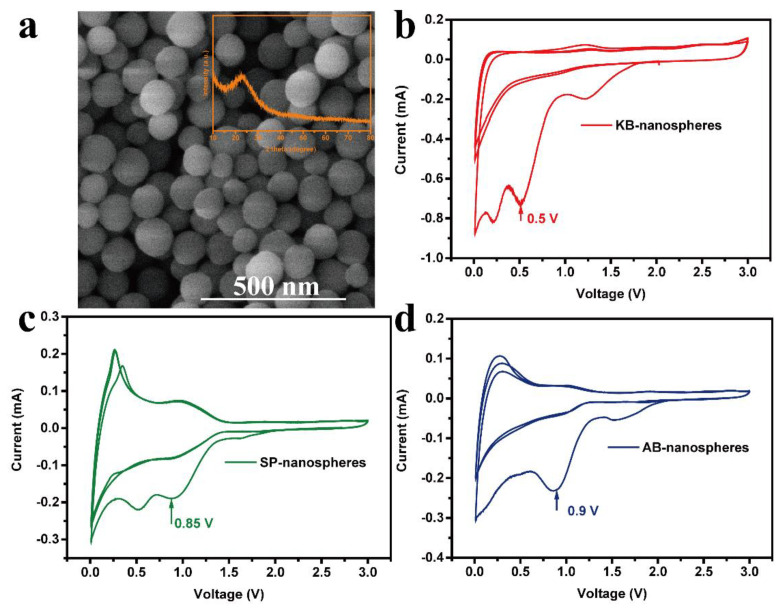
The SEM image and XRD pattern (the inset) (**a**) of silica nanospheres, CV curves of the first three cycles of the silica nanospheres when using KB (**b**), SP (**c**), and AB (**d**) as conductive agents, respectively.

**Figure 10 nanomaterials-12-00692-f010:**
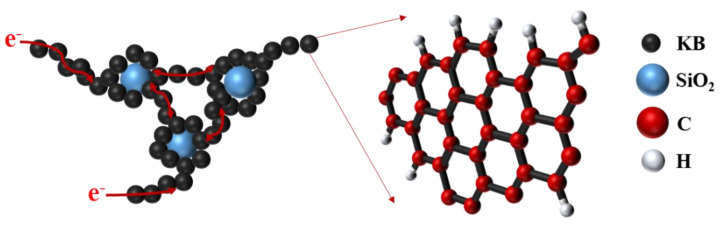
Schematic diagram conductive network of KB.

**Table 1 nanomaterials-12-00692-t001:** The electrical resistance and electrical resistivity of anode electrodes.

Sample	SP	AB	KB
sheet resistance (Ω)	125.0	148.2	64.9
electrical resistivity (Ω·cm)	12.50	14.82	6.49

## Data Availability

The data presented in this study are available from the corresponding author upon request.

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
