# Peer review of "Improvement of Lithium Storage Performance of Silica Anode by Using Ketjen Black as Functional Conductive Agent"

_nanomaterials, 2022, doi:10.3390/nano12040692_

Round 1

Reviewer 1 Report

Authors report on the use of different C materials to be integrated in Si based elecrodes, the work is clear and convincing, I suggest this minor revisions before acceptance:

(1) Please define ec/dmc/dec acronyms 

(2) "KB has the best dispersion, indicating it has the best electrochemical properties (tem)”. Pretending a direct extrapolation of EC properties from TEM is a little too simplistic! (reformulate)

(3) Please revise figure 3 with more clear legend, I suggest also to format it with 4 panels with the same size.

(4) For XPS spectra, please precise how the energy calibration was performed and if charge compensation was used (flood gun). Could you observe on C1s or O1s spectra hints of formation of chemical bonds between SiO2 and Carbon (i.e. Si-O-C-H bonds)

(5) “A longer period of decrease in the diffusion rate of Li+ in SiO2-KB is since Li+ diffuses from the SiO2 inner core to the electrolyte over a longer distance from the SiO2 surface to the electrolyte than using the other two conductive agents.” Confuse, please reformulate this.

(6) Suggested reference on stability and performance of C support:
10.1021/acscatal.7b03539

Author Response

  1.   Please define ec/dmc/dec acronyms.

Answer: Thanks for the invaluable comments. We have added the explanation in the revised manuscript. (Please see Electrochemical measurements as highlighted in red, page 2)
2.  "KB has the best dispersion, indicating it has the best electrochemical properties (tem)”. Pretending a direct extrapolation of EC properties from TEM is a little too simplistic! (reformulate)

Answer: Correction has been made in the revised manuscript. Please see page 3 as highlighted in red.
3.  Please revise figure 3 with more clear legend, I suggest also to format it with 4 panels with the same size.

Answer: Thank you for pointing out this issue. We have revised figure 3 in the revised manuscript, page 5.
4.  For XPS spectra, please precise how the energy calibration was performed and if charge compensation was used (flood gun). Could you observe on C1s or O1s spectra hints of formation of chemical bonds between SiO2 and Carbon (i.e. Si-O-C-H bonds)

Answer: Thanks for the invaluable comments. Energy calibration for surface contamination was performed using C1s (284.6eV) as a standard. We cannot confirm the Si-O-C-H bond, but based on studies of the reaction mechanism of this SiO2 anode over the years, there should be no Si-O-C-H bond.
5. “A longer period of decrease in the diffusion rate of Li+ in SiO2-KB is since Li+ diffuses from the SiO2 inner core to the electrolyte over a longer distance from the SiO2 surface to the electrolyte than using the other two conductive agents.” Confuse, please reformulate this.

Answer: Thank you for pointing out this issue. Correction has been made in the revised manuscript. Please see page 8 as highlighted in red.

  1. Suggested reference on stability and performance of C support: 10.1021/acscatal.7b03539

Answer: Thanks for the invaluable comments. We have quoted this reference in the revised manuscript. (Please see reference 30)

Reviewer 2 Report

The manuscript ‘Improvement of lithium storage performance of silica anode by using ketjen black as functional conductive agent’ reports the comparison of electrochemical performances of SiO2 composited with three different carbon additives: KB, SP, and AB. The results are noteworthy and appear relevant for publication in Nanomaterials. The material and electrochemical characterization in the manuscript appear comprehensive and seem to support the claims made. However, there are a few minor concerns, especially with the electrochemical characterization, which would need to be addressed before I can recommend this manuscript for publication in Nanomaterials. They are listed as follows:

  1. In the Introduction, there seems to have been a slightly abrupt jump from discussing silicon (Si) to discussing silica (SiO2) composites. It would be good if the authors briefly discuss the motivation behind using SiO2 as anode material, key differences between Si and SiO2 during cycling, etc. before discussing the SiO2 composites for LIB applications
  2. In Section 2.3, the authors mention that they used 100 mA/g as the current rate for their electrochemical measurements. Is this with respect to the weight of SiO2 or the weight of the composite i.e., SiO2+KB/SP/AB? This distinction is important.
  3. Few more details on the electrochemical methods would be useful for researchers trying to reproduce/follow-up on the experiments. Specifically: What was the volume of the electrolyte used in the half cells? What was the thickness of the Li foil? What was the separator used?
  4. In Figure 3a, the specific capacity of SiO2-KB seems to decrease until ~40th cycle and then gradually increases even beyond 100 cycles. This phenomenon seems to happen only with KB and not with SP or AB composites. Could the authors hypothesize as to why this is happening?
  5. In Figure 3a, the right y-axis (Coulombic efficiency) could be zoomed in to provide more meaningful insight into the differences in the CE, perhaps between 80-100%.
  6. In Figure 3c, I would encourage the authors to include the equivalent circuit used to fit the Nyquist plots.
  7. In Figure 3d, what is the reason behind the specific capacity of SiO2-KB fluctuating between 200 to 400 cycles?

Author Response

  1. In the Introduction, there seems to have been a slightly abrupt jump from discussing silicon (Si) to discussing silica (SiO2) composites. It would be good if the authors briefly discuss the motivation behind using SiO2 as anode material, key differences between Si and SiO2 during cycling, etc. before discussing the SiO2 composites for LIB applications

Answer: Thanks for the invaluable comments. The introduction has been revised and added the previous research summary in the revised manuscript. Please see in introduction (page 1) as highlighted in red.

  1. In Section 2.3, the authors mention that they used 100 mA/g as the current rate for their electrochemical measurements. Is this with respect to the weight of SiO2 or the weight of the composite i.e., SiO2+KB/SP/AB? This distinction is important.

Answer: Thank you for pointing out this issue. The current density is calculated by the weight of SiO2. Correction has been made in the revised manuscript, please see page 2 as highlighted in red.

  1. Few more details on the electrochemical methods would be useful for researchers trying to reproduce/follow-up on the experiments. Specifically: What was the volume of the electrolyte used in the half cells? What was the thickness of the Li foil? What was the separator used?

Answer: Thank you for pointing out this issue. We have added more details on the electrochemical methods  in the revised manuscript. Please see page 2 as highlighted in red.

  1. In Figure 3a, the specific capacity of SiO2-KB seems to decrease until ~40th cycle and then gradually increases even beyond 100 cycles. This phenomenon seems to happen only with KB and not with SP or AB composites. Could the authors hypothesize as to why this is happening?

Answer: Thank you for pointing out this issue. When using AB and SP as conductive agents, this phenomenon is not evident due to the low capacity, but the similar trend of falling first and then rising can be seen in the high current density cycle, as shown in Figure 3d. The reason has been explained in page 4 as highlighted in red in the revised manuscript.

  1. In Figure 3a, the right y-axis (Coulombic efficiency) could be zoomed in to provide more meaningful insight into the differences in the CE, perhaps between 80-100%.

Answer: Thank you for pointing out this issue. However, we think this is the best solution for an aesthetically pleasing illustration.

  1. In Figure 3c, I would encourage the authors to include the equivalent circuit used to fit the Nyquist plots.

Answer: Thanks for the invaluable comments. We have added the Nyquist plots in figure 3c and provided the explanation in page 4 as highlighted in red  in the revised manuscript.

  1. In Figure 3d, what is the reason behind the specific capacity of SiO2-KB fluctuating between 200 to 400 cycles?

Answer: Thanks for the invaluable comments. This is the evidence of the gradual activation reaction of silica to form silicon and silicon by-products.

Reviewer 3 Report

Thanks for the invitation to review the manuscript which showed the effect of conductive carbon on the performance of LIB where SiO2 was prepared as anode material. The manuscript can be published but require major revision

  1. Introduction can be improved by including few lines about other anodes and their disadvantage. Moroever, authors can also include various reaction mechanism involved during charge discharge.
  2. The whole manuscript discuss about three kind of carbon but as a reader i could not see why KB works better. Authors should summarise their work better.
  3. Does surface area or pore size have also any impact? KB is known as highly conductive carbon and considered as not cost-effective approach. Also, does it solve the probelm of volume expansion.
  4. Why authors called it aerogel. It looks more of nanospehers.
  5. Important refernces should be included Energy Environ. Sci., 2012,5, 6895-6899; https://doi.org/10.1002/smll.202006651; J. Mater. Chem. A, 2015, 3, 22739.

Author Response

  1. Introduction can be improved by including few lines about other anodes and their disadvantage. Moroever, authors can also include various reaction mechanism involved during charge discharge.

Answer: Thank you for pointing out this issue. We have made improvement in the introduction and added various reaction mechanisms involved during charge and discharge in page 6 as highlighted in red in the revised manuscript.

  1. The whole manuscript discuss about three kind of carbon but as a reader i could not see why KB works better. Authors should summarise their work better.

Answer: Thanks for the invaluable comments. We have recapitulated the conclusions in the revised manuscript. Please see page 9 as highlighted in red.

  1. Does surface area or pore size have also any impact? KB is known as highly conductive carbon and considered as not cost-effective approach. Also, does it solve the problem of volume expansion?

Answer: Thanks for the invaluable comments. The surface area or pore size may have some effect on the properties and we will discuss it in detail in the next step. From the morphological evolution of SiO2 electrode in Figure 4, it is clear that KB has a very obvious effect of inhibiting the volume expansion of silica.

  1. Why authors called it aerogel. It looks more of nanospehers.

Answer: Thank you for pointing out. We called it aerogel because the silica is prepared by freeze-drying from gel.

  1. Important refernces should be included Energy Environ. Sci., 2012,5, 6895-6899; https://doi.org/10.1002/smll.202006651; J. Mater. Chem. A, 2015, 3, 22739.

Answer: Thanks for the invaluable comments. We have quoted those references in the revised manuscript. (Please see reference 1, 17, 28, 46 and 47.)

Reviewer 4 Report

In this paper, the authors investigate the electrochemical performance of SiO2 gel anode with Ketjen Black (KB), Super P (SP), and Acetylene Black (AB) as conductive agents. They demonstrate that Ketjen Black (KB) provides better cycling and rate performance than SP and AB for SiO2 anode electrodes.

The paper is based on a seemingly well-conducted experimental work and offers data that can be of interest to the scientific community.

The paper could be suitable for publication after a revision.

Here are suggestions and remarks for the authors:

  1. Improve English grammar and style.
  2. “Compared with AB and SP, Ketjen Black (KB) has the advantages of large specific surface area, excellent electrical conductivity, and relatively narrow pore size distribution, when used as the conductive agent[30].” Although references are given, the authors could add something more about the chemical and physical properties of AB, SP, and KB, respectively.
  3. Figure 2: if possible, I suggest showing the three TEM images with the same scale, i.e at the same magnification.
  4. Give the sheet resistance in Ohm/square
  5. "The rate capability of the electrodes is shown in Figure 3b, with different current densities of 0.2, 0.4, 0.6, 0.8, 1.0, 2.0, and 4.0 A g-1. The SiO2-KB electrode attains an average discharge capacity of 318.2 mA h g-1, 263.6 mA h g-1, 231.9 mA h g-1, 213.4 mA h g-1, 207.7 mA h g-1, 176.7 mA h g-1, and 151.4 mA h g-1 respectively at the above current densities." Here and elsewhere, if evident from the plots, avoid giving numerical details that make the reading heavy and distracting.
  6. Axes labels and numbers are difficult to read in several plots. I suggest using a higher font size. Also, for double-y plots, specify which curve refers to which axis, although it might be obvious.
  7. “Figure 1 Schematic diagram conductive network of KB”. It should be figure 10.

Author Response

  1. Improve English grammar and style.

Answer: Thanks for the noteworthy comments. The manuscript has been carefully revised considering about the language and grammar problems.

  1. “Compared with AB and SP, Ketjen Black (KB) has the advantages of large specific surface area, excellent electrical conductivity, and relatively narrow pore size distribution, when used as the conductive agent[30].” Although references are given, the authors could add something more about the chemical and physical properties of AB, SP, and KB, respectively.

Answer: Thanks for the invaluable comments. The main aim of the manuscript is to investigate the different electrochemical reactions of silica anode using three conductive agents, and therefore the characterization has been based on composite materials. Furthermore, the KB, SP and AB chemical and physical properties have been characterized in the cited article, which we have referred to when designing the experiments, and we have also performed experiments with consistent experimental data, which has not been added to this manuscript for the sake of the flow of the article.

  1. Figure 2: if possible, I suggest showing the three TEM images with the same scale, i.e at the same magnification.

Answer: Thanks for the suggestion. We have chosen the most appropriate size to show the picture.

  1. Give the sheet resistance in Ohm/square

Answer: Thanks for the suggestion. The resistivity of the four probes we tested is in ohm*cm.

  1. "The rate capability of the electrodes is shown in Figure 3b, with different current densities of 0.2, 0.4, 0.6, 0.8, 1.0, 2.0, and 4.0 A g-1. The SiO2-KB electrode attains an average discharge capacity of 318.2 mA h g-1, 263.6 mA h g-1, 231.9 mA h g-1, 213.4 mA h g-1, 207.7 mA h g-1, 176.7 mA h g-1, and 151.4 mA h g-1 respectively at the above current densities." Here and elsewhere, if evident from the plots, avoid giving numerical details that make the reading heavy and distracting.

Answer: Thanks for the suggestion. We have minimized the numerical details of SiO2-SP and SiO2-AB.

  1. Axes labels and numbers are difficult to read in several plots. I suggest using a higher font size. Also, for double-y plots, specify which curve refers to which axis, although it might be obvious

Answer: Thanks for the invaluable comments. We have made the correction.

  1. “Figure 1 Schematic diagram conductive network of KB”. It should be figure 10.

Answer: Thank you for pointing out. We have corrected this error in the revised manuscript.

Round 2

Reviewer 3 Report

Authors have revised manuscript well as per reviewers remarks therefore it can be published in its current form.